# Intraoperative Near-Infrared Fluorescence Guided Surgery Using Indocyanine Green (ICG) for the Resection of Sarcomas May Reduce the Positive Margin Rate: An Extended Case Series

**DOI:** 10.3390/cancers13246284

**Published:** 2021-12-14

**Authors:** Marcus J. Brookes, Corey D. Chan, Fabio Nicoli, Timothy P. Crowley, Kanishka M. Ghosh, Thomas Beckingsale, Daniel Saleh, Petra Dildey, Sanjay Gupta, Maniram Ragbir, Kenneth S. Rankin

**Affiliations:** 1North of England Bone and Soft Tissue Tumour Service, Royal Victoria Infirmary, Queen Victoria Road, Newcastle upon Tyne NE1 4LP, UK; corey.chan@newcastle.ac.uk (C.D.C.); fabio.nicoli@northumbria-healthcare.nhs.uk (F.N.); timothy.crowley1@nhs.net (T.P.C.); kanishka.ghosh1@nhs.net (K.M.G.); t.beckingsale@nhs.net (T.B.); daniel.saleh2@nhs.net (D.S.); petra.dildey@nhs.net (P.D.); maniram.ragbir@nhs.net (M.R.); 2Translational and Clinical Research Institute, Newcastle University, Newcastle upon Tyne NE1 7RU, UK; 3Department of Musculoskeletal Oncology Surgery, Glasgow Royal Infirmary, 84 Castle St., Glasgow G4 0SF, UK; sanjay.gupta@ggc.scot.nhs.uk

**Keywords:** indocyanine green, ICG, sarcoma, near-infrared fluorescence, NIR, margin

## Abstract

**Simple Summary:**

Sarcomas are rare cancers that can arise all over the body, with many occurring in younger people. Treatment is usually based around surgery to remove the tumour. It is crucial to remove all of the tumour to minimise the chance of the tumour coming back. Currently, surgeons review scans of the tumour and plan the operation beforehand, but it can be difficult to relate the scans to what the surgeon sees during the operation. Currently, there are no established technologies to help them identify the tumour during the operation. Here we have given patients a harmless dye (indocyanine green) prior to the operation, which makes the tumour glow on a screen when seen by a camera during the operation, helping the surgeon to identify and remove the tumour. This resulted in the surgeon being able to remove all of the tumour more often, reducing the chances of it coming back.

**Abstract:**

Background: Sarcomas are rare, aggressive cancers which can occur in any region of the body. Surgery is usually the cornerstone of curative treatment, with negative surgical margins associated with decreased local recurrence and improved overall survival. Indocyanine green (ICG) is a fluorescent dye which accumulates in sarcoma tissue and can be imaged intraoperatively using handheld near-infrared (NIR) cameras, theoretically helping guide the surgeon’s resection margins. Methods: Patients operated on between 20 February 2019 and 20 October 2021 for intermediate to high grade sarcomas at our centres received either conventional surgery, or were administered ICG pre-operatively followed by intra-operative NIR fluorescence guidance during the procedure. Differences between the unexpected positive margin rates were compared. Results: 115 suitable patients were identified, of which 39 received ICG + NIR fluorescence guided surgery, and 76 received conventional surgery. Of the patients given ICG, 37/39 tumours fluoresced, and surgeons felt the procedure was guided by the intra-operative images in 11 cases. Patients receiving ICG had a lower unexpected positive margin rate (5.1% vs. 25.0%, *p* = 0.01). Conclusions: The use of NIR fluorescence cameras in combination with ICG may reduce the unexpected positive margin rate for high grade sarcomas. A prospective, multi-centre randomised control trial is now needed to validate these results.

## 1. Introduction

Sarcomas are rare (~1% of all cancers [1]) but typically aggressive malignant tumours arising from mesenchymal tissues, with 60% arising in the extremities [2]. Sarcomas are a heterogenous group of tumours, comprising over 70 subtypes with varied characteristics [3]. Typically, the mainstay of curative treatment for localised high grade sarcomas is a combination of surgery and radiotherapy to the primary site, delivered in either the neoadjuvant or adjuvant setting, although this varies between subtypes and centres [4]. Surgically, the most important factor in terms of oncological outcome is the resection margin. Large-scale retrospective analyses have shown positive microscopic margins to be associated with both an increased rate of local recurrence (LR) [5] and reduced overall survival (OS) [5,6,7], whilst a recent meta-analysis also found decreased OS with a positive microscopic margin [8]. Despite this, little advancement has been made in recent years to reduce the positive margin rate which remains considerable, with quoted incidence varying between 8 and 29.9% in larger studies [5,6,9,10,11,12,13]; the positive margin rate is often higher in more infiltrative subtypes such as myxofibrosarcoma [14].

In 2019, our group published the first case series, containing 11 patients, demonstrating near-infrared (NIR) fluorescence guidance for open sarcoma surgery using Indocyanine Green (ICG) [15]. ICG is a tricarbocyanine dye [16] with an established safety profile demonstrating excellent safety characteristics, with the exception of those allergic to iodine, who are at risk of anaphylaxis [17,18]. When given intravenously (IV) prior to surgery, ICG accumulates mainly in the tumour, and less so in surrounding normal tissue, causing the tumour to fluoresce when viewed with an NIR camera [15]. The exact mechanism of preferential accumulation of ICG within tumour tissue for sarcomas is not yet certain, although it is thought to be a combination of enhanced uptake secondary to upregulated clathrin mediated endocytosis by tumour cells [19] and the enhanced permeability and retention (EPR) effect [20,21,22]; this effect describes the accumulation of ICG within the tumour due to the aberrant, leaky vasculature combined with increased retention secondary to impaired lymphatic drainage. The aim of this is to help the surgeon identify the tumour intra-operatively to help guide resection, hopefully reducing the positive margin rate. The use of ICG to guide oncological resection has previously been described in multiple cancers [23,24,25]; with a recently published study reporting that the lack of residual fluorescence in the tumour bed was predictive of a negative surgical margin in breast cancer [26].

In our original case series, we demonstrated that ICG successfully made the tumour fluoresce under NIR, and that this guided the surgeon with regards to their margin in 3/11 cases [15]. Having now completed many more cases, we are able to update our findings, having answered more questions about the practicalities of NIR guided surgery, including the dosage and timing of ICG administration, as well as patient factors that affect suitability for ICG guided sarcoma resection. Furthermore, we are now able to compare the positive margin rates between patients receiving NIR guided surgery and those receiving conventional treatment, to offer an early indication as to whether ICG can reduce the positive margin rate as hypothesised.

## 2. Materials and Methods

### 2.1. Patients

Patients who underwent definitive surgery for high grade sarcomas at the North of England Bone and Soft Tissue Tumour Service between 20 February 2019 and 20 October 2021 and under the care of the Glasgow Sarcoma Service were identified from prospectively maintained databases. Standard practice at the North of England Bone and Soft Tissue Tumour Service is to deliver radiotherapy post-operatively in most cases whereas at the Glasgow Sarcoma Service, decisions on radiotherapy timing are made on a case-by-case basis, influenced by grade, size, depth, subtype, anatomical location and proximity to critical structures. Patients with visceral, head and neck or retro-peritoneal sarcomas were excluded, as were patients with recurrent sarcomas or patients in which the margin status was unknown.

Patient’s electronic records were reviewed, and the following information collected: Age at diagnosis, gender, tissue type (bone or soft tissue), site of tumour, size of tumour, histological subtype, grade of tumour, type of surgery (wide excision or amputation), closest microscopic margin, wound complications (including any dehiscence, infection, necrosis or seroma requiring intervention), consultants involved in the surgery and neoadjuvant therapies. Margins were defined according to the pathology report, classifying margins as positive if tumour cells were visualised at the microscopic margin (0 mm), in line with the R classification system [27]. Operation notes were reviewed to clarify whether these positive margins were expected (usually due to the preservation of critical structures) or unexpected positive margins (UPM). Grading and classification of the sarcomas was carried out by experienced sarcoma pathologists, in accordance with WHO guidance and in line with the French Fédération Nationale des Centres de Lutte Contre le Cancer (FNCLCC) grading system [28], with grades 2 and 3 classified as high grade. As the FNCLCC grading system does not apply to all sarcoma subtypes, sarcomas reported as of a morphologically high grade in the pathology report were also included. Tumour size was recorded from the histology report, or the MRI report if this was not available. Tumour depth was taken from the MRI report, according to its relationship to the fascia.

### 2.2. Surgery

Patients were either treated with conventional surgery (conventional surgery, i.e., no NIR fluorescence guidance) or were given ICG pre-operatively followed by NIR fluorescence guidance. This decision was not randomised, but was due to the availability of both the camera system and the dye, as well as an available bed to admit the patient the day before. Initially, most NIR fluorescence procedures were carried out by one surgeon; this was adopted by the others through the course of this study. Timing of administration was either the afternoon prior to surgery (16–24 h prior to surgery) or at induction of anaesthesia, both via the IV route. ICG dosing varied, with fixed doses of 25–100 mg and weight-based dosing of 1 mg/kg trialled; all doses were well below the recommended maximum dose of 2 mg/kg. If ICG was given, assessments of tumour fluorescence were made intra-operatively using the Stryker Spy Phi near-infrared camera (Stryker Corp., Kalamazoo, MI, USA). If ICG was being used, the skin was prepared using chlorhexidine to prevent background fluorescence caused by iodine preparations. The surgeon would then mark up the skin as normal, as per imaging and/or palpation. They would then proceed to image the area, taking photos on all four camera modes (white light, SPY mode, overlay mode and colour-segmented mode). They would then proceed with the operation, repeating the imaging process as above following dissection through the skin and fat (and then following dissection of the deeper layers if required). Following resection of the specimen, the imaging process was repeated for both the resected specimen on a side table, and the wound bed. Prior to imaging, the operating lights would be directed away from the operating field to reduce background fluorescence. At any imaging point, the surgeon used their judgement to decide whether to alter their operative procedure according to the images; immediately following the operation, they were asked whether the images guided the surgery and, if so, how they guided it. This was approved by the local institutional review board (Caldicott number 7159).

### 2.3. Statistical Analysis

Differences between patient groups were assessed using Pearson’s chi-squared or Fisher’s exact test depending on group sizes, whilst the difference in positive margin rates between groups was assessed with Fisher’s exact test and binary logistic regression was used to create univariate and multivariate models. The following variables were included in the multivariate analysis as they had previously been implicated with either increased positive margin rates or increased LR: size (<50 mm or ≥50 mm) [5,29], depth relative to the fascia [12], invasive subtype (myxofibrosarcoma or undifferentiated pleomorphic sarcoma (UPS)) [11,30] and site (extremity or non-extremity) [12]. All statistical analysis was performed using SPSS (version 27, IBM, Chicago, IL, USA).

## 3. Results

### 3.1. Patient Characteristics

Patient characteristics and basic clinical information are displayed in Table 1; 115 patients in total were identified as suitable, with 39 patients having received ICG pre-operatively and 76 receiving conventional surgery. The *p* values demonstrate the probability of a significant difference in the distribution of subcategories between treatment groups. Given the non-randomised nature of the methodology, there were a number of significant differences between groups, notably in primary tumour locations, sarcoma subtypes and neoadjuvant therapies received. The ICG group contained higher percentages of invasive subtypes (myxofibrosarcoma and UPS) pelvic tumours and larger tumours, whilst containing fewer patients receiving neoadjuvant chemotherapy and patients < 25 years of age.

### 3.2. Fluorescence

Table 2 displays more in-depth information about all cases in which ICG was used for NIR fluorescence guidance, including ICG doses, administration time and neoadjuvant therapies. Out of 39 cases, 37 fluoresced under the NIR camera; of the two cases that did not fluoresce, one was a patient with a grade 2 myxofibrosarcoma to whom only 25 mg of ICG was administered, whilst the other was a patient with tibial osteosarcoma who had undergone neo-adjuvant chemotherapy with >90% necrosis. One patient with a large grade 2 myxofibrosarcoma received neoadjuvant radiotherapy at the Glasgow Sarcoma Service due to the size, depth and anatomical location of the tumour and the tumour did fluoresce. In 11/39 cases, the surgeon felt that the ICG fluorescence helped guide their intra-operative decision making with regards to margins taken. NIR fluorescence guided resections were carried out by eight different surgeons; the technology was found to be easy to use and quick to learn.

### 3.3. Resection Margins

The surgical margins achieved in both groups are displayed in Table 3. Overall, the cohort had an UPM rate of 22.3% across both treatment groups. The UPM margin rate was significantly reduced in the group receiving ICG when compared to those receiving conventional surgery (5.1% vs. 25.0%, *p* = 0.010), with a relative odds ratio of 0.162 (*p* = 0.019).

This retains significance when tumour size (<50 mm vs. ≥50 mm), site (extremity vs-non-extremity) and invasive phenotype (myxofibrosarcoma/UPS vs. other sarcomas), are included as a multivariate model (*p* = 0.007) (Table 4); all variables were significant predictors of UPM in this model (size *p* = 0.046, site *p* = 0.041, invasive phenotype *p* = 0.001). After MDT discussion, 2/39 (5.1%) patients in the ICG group underwent re-excision of the wound bed whilst 10/76 (13.2%) patients receiving conventional treatment underwent further excision; there was no significant difference in the re-excision rate however (*p* = 0.218).

### 3.4. Dosage and Administration Time of ICG

Administration of ICG at the induction of anaesthesia was trialled in three cases, but it was felt this resulted in increased background fluorescence, thereby reducing the utility of ICG for surgical guidance; this is demonstrated by Figure 1, whilst Figure 2 shows a patient in which the ICG was administered the day before surgery, with improved specificity between tissue types clearly visible.

### 3.5. Wound Complications

In the immediate postoperative period, 15/39 (38.5%) had a wound complication, compared to 25/76 (32.9%) of those receiving standard surgical care; this was not significant (*p* = 0.68). Most complications were minor, with complete resolution reached prior to discharge.

## 4. Discussion

Having already demonstrated the feasibility of NIR fluorescence guidance using ICG for sarcoma surgery [15], we believe this study now provides early evidence that it may reduce the positive margin rate in high grade sarcoma resection, with a UPM of 5.1% in the NIR fluorescence guided group compared to 25.0% in the group receiving conventional surgery; this is despite the ICG group containing higher percentages of invasive soft tissue sarcomas, which typically have higher positive margin rates [14], as well as larger tumours. Retaining significance at the multivariate level also helps strengthen this finding. Furthermore, this may be of relevance to other solid cancers, given the use of ICG has already been described for the resection of multiple sub-types; this is the first paper we are aware of however to directly compare the positive margin rate when using ICG compared with conventional surgery. However, in the absence of randomisation, and with its retrospective nature, this evidence is not conclusive, and requires confirmation in the form of a multi-centre randomised control trial; this is particularly important given the heterogenous nature of sarcomas meaning a large sample is required in order to ensure it is fully representative.

A number of other particularly useful findings have also come out of this study. The optimal time of administration of ICG was not yet known; previous papers using ICG for other cancer types have given the ICG at the induction of anaesthesia (for breast cancer resection) [26], after anaesthesia once the abdomen was open (for ovarian cancer) [23] or the day before surgery (for pulmonary metastasectomy) [31]. After trialling ICG administration at both the induction of anaesthesia and the day before, we feel the day before is optimal for tumour tissue identification, as there was felt to be less non-specific background fluorescence. This may relate in part to the differing mechanism of ICG uptake and clearance in various cell types. Studies using a mouse model of colorectal cancer suggested no specific tumour delivery, but preferential uptake of ICG in the tumour tissue secondary to increased clathrin mediated endocytosis rates in cancer cells [19]. They also demonstrated retention of ICG by cancer cells for a minimum of 24 h; contrarily ICG was cleared rapidly from normal tissues [19]. Other research has suggested that ICG accumulates in the tumours due to the enhanced permeability and retention (EPR) effect, aided by the dense but leaky vasculature within tumours, coupled with impaired lymphatic drainage, leading to increased ICG retention [20,21,22]. Recent basic science research by our team using sarcoma cell lines supports both theories simultaneously, demonstrating clathrin mediated endocytosis as a key mechanism for the uptake of ICG, whilst increased retention was likely dependent on the EPR effect, where ICG pools within the tumour due to the aberrant architecture [32]. These studies support our clinical findings of improved specificity when administering ICG the day prior to surgery—ICG is retained by the tumour for at least 24 h, but this ensures sufficient time for it to be cleared from surrounding tissues. Reported dosing of ICG has also varied widely in the literature, from doses of 0.25 mg/kg [26] to 5 mg/kg [31] (recommended maximum dose of 2 mg/kg); we feel confident that a dose of 1 mg/kg provided adequate tumour fluorescence with minimal background fluorescence. A dose of 75 mg of ICG was used initially; 25 mg was trialled in one patient but, after this failed to fluoresce, we reverted to 75 mg. Two patients received 100 mg; this was due to their above average body mass. Towards the end of this study, we switched to a weight-based dose (1 mg/kg) in order to standardise dosing in preparation for a prospective randomised control trial. Whilst we are satisfied with the results using the current dose of 1 mg/kg, and it is significantly below the maximum dose of 2 mg/kg, we may be able to reduce this in future given the lower doses used in other studies [23,24,26,33]. Furthermore, we demonstrated that ICG guided surgery may be feasible in patients receiving radiotherapy in the neoadjuvant setting, which is particularly important given this is still the preferred practice in many centres around the world; however, further cases will need to be performed in order to confirm this.

Whilst we feel that the use of ICG is beneficial, fluorescence guided surgery is still a young, developing technology. Although non-specific fluorescence can be minimised via the optimisation of dosing regimens, it remains a non-targeted dye and admitting patients the day prior to surgery to administer the dye has obvious cost implications. Secondly, the extent of non-specific fluorescence using a non-targeted dye such as ICG is difficult to predict, and will vary greatly between cases given the heterogeneity of sarcoma tumours and the large variety in patient phenotypes. Non-specificity may in future be overcome by the binding of fluorophores to monoclonal antibodies specific to cell surface markers overexpressed by cancer cells. Sardar et al. recently demonstrated that the use of a fluorophore-bound antibody and ICG simultaneously improved the quality of imaging in a mouse synovial sarcoma model, with ICG accumulating in the necrotic areas of the tumour, whilst the dye bound to the monoclonal antibody accumulated in the more cellular regions of the tumour [34]. Whilst targeted dyes are one option to improve specificity, Cahill et al. have recently described the use of artificial intelligence (AI) to train computer software to differentiate between background fluorescence and tumour fluorescence when ICG is administered to patients with colorectal cancer intra-operatively and dynamic studies of normal versus tumour tissue fluorescence are performed [35]. This technology, in theory, would improve the specificity of ICG in sarcoma resection, whilst quantifying fluorescence would allow an objective measure as opposed to the subjective opinion of the surgeon reviewing the images intra-operatively.

The UPM rate of 25% reported in this study for those receiving conventional surgery is on the higher side of those reported in the literature. We believe there are a number of reasons for this, including the preference at the North of England Bone and Soft Tissue Tumour Service for post-operative radiotherapy which may allow for clearer identification of sarcoma cells at the margins. The other factor, again at the North of England Bone and Soft Tissue Tumour Service, is the staged approach taken for invasive soft tissue sarcomas, in which wounds are not closed until the histological margins are reported, allowing further resection prior to definitive closure if necessary; this has been shown to result in low rates of local recurrence despite high initial positive margin rates [36]. When excluding these invasive soft tissue sarcomas in which this technique is used from this series, the UPM rate dropped to 11.1%, which sits at the lower end of those reported in the literature [5,6,9,10,11,12,13].

One concern with this technology is that, given the non-specific nature of ICG, it may lead to the over-resection of tissue due to concerns caused by non-specific fluorescence of normal tissue. The immediate concern would be that of increased wound complications; whilst the rate was slightly higher in the ICG group here, this was not significant. In the longer term, the problem with over-resection would be the potential detriment to patient’s functional status. This is something we have not assessed, but functional outcomes and patient reported outcome measures must be reported in future, prospective studies.

Whilst these results suggest that NIR fluorescence guided surgery with ICG reduces the UPM rate in high-grade sarcoma resections, this may not be due to a direct causative effect; it could be that the use of this technology forces surgeons to plan their resections in further detail, which in turn leads to the reduced UPM rate reported in this study.

Despite the limitations of this study, namely its non-randomised and retrospective nature, we believe this paper provides evidence that the administration of 1 mg/kg of ICG 16–24 h prior to surgery, in combination with the use of a NIR camera intra-operatively, may reduce the UPM rate in high grade sarcomas. This is the first study to demonstrate a reduction in the UPM in soft-tissue sarcomas via the use of novel surgical technique and as such should be followed up with a multi-centre, randomised control trial in order to formally evaluate this technology. Integration of AI to NIR camera systems to help improve the specificity of ICG fluorescence, as well as the development of sarcoma-specific monoclonal antibody conjugates, must happen simultaneously in order to further improve this promising and exciting technology.

## 5. Conclusions

This study demonstrates that, not only is the use of ICG for NIR fluorescence guided surgery feasible in high grade sarcomas, but that it may reduce the unexpected positive margin rate. Further study, in the form of a multi-centre, randomised control trial is required to validate this technology. Work should also be carried out to incorporate AI technology into handheld NIR cameras and to develop specific, sarcoma targeted dyes in order to improve the specificity of this technology.

## Figures and Tables

**Figure 1 cancers-13-06284-f001:**
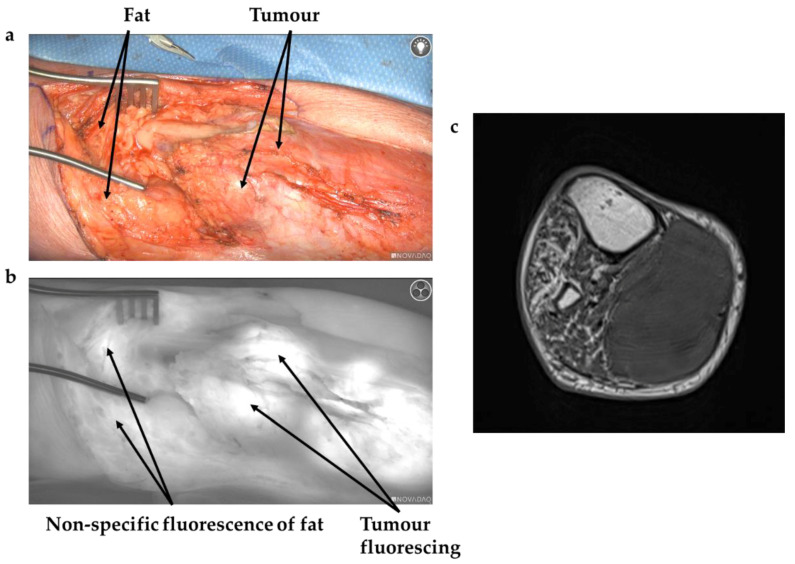
(**a**) White light image of approach to leiomyosarcoma in gastrocnemius (**b**) Demonstrates the same view imaged with NIR camera after ICG given at induction. Arrows demonstrate background fluorescence of normal tissue. (**c**) Axial MRI of tumour.

**Figure 2 cancers-13-06284-f002:**
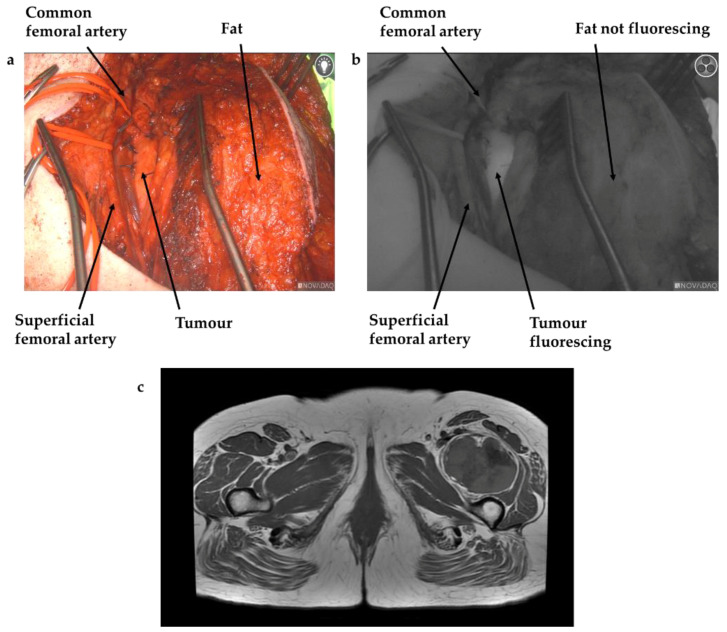
(**a**) White light image of approach to synovial sarcoma in thigh, with anatomical landmarks labelled. (**b**) Demonstrates the same view imaged with NIR camera after ICG administered 16–24 h pre-operatively, with the tumour glowing brightly, and minimal background fluorescence in normal tissues. (**c**) Axial MRI of tumour.

**Table 1 cancers-13-06284-t001:** Patient demographics and clinical information.

Characteristic	Conventional, *n* (%)	ICG-Guided, *n* (%)	*p*-Value
**Age, years**			
<25	18 (23.7%)	1 (2.6%)	
25–50	12 (15.8%)	5 (12.8%)	
>50	46 (60.5%)	33 (84.6%)	*p* = 0.005 *
**Gender**			
Male	49 (64.5%)	24 (61.5%)	
Female	27 (35.5%)	15 (38.5%)	*p* = 0.839
**Primary location**			
Upper extremity	8 (10.5%)	9 (23.1%)	
Lower extremity	46 (60.5%)	21 (53.8%)	
Pelvis	0 (0.0%)	6 (15.4%)	
Trunk	17 (22.4%)	2 (5.1%)	
Groin	5 (6.6%)	1 (2.6%)	*p* < 0.001 *
**Tumour depth**			
Superficial	26 (34.2%)	11 (28.2%)	
Deep	50 (65.8%)	28 (71.8%)	*p* = 0.536
**Tumour size**			
<50 mm	18 (23.7%)	4 (10.3%)	
≥50 mm	58 (76.3%)	35 (89.3%)	*p* = 0.131
**Tissue type**			
Bone	24 (31.6%)	13 (33.3%)	
Soft tissue	52 (68.4%)	26 (66.7%)	*p* = 0.837
**Histology**			
Myxofibrosarcoma	15 (19.7%)	10 (25.6%)	
UPS	16 (21.1%)	9 (23.1%)	
Chondrosarcoma	5 (6.6%)	8 (20.5%)	
Liposarcoma	8 (10.5%)	0 (0.0%)	
Synovial sarcoma	2 (2.6%)	3 (7.7%)	
Ewing sarcoma	7 (9.2%)	0 (0.0%)	
Osteosarcoma	11 (14.5%)	2 (5.1%)	
Leiomyosarcoma	4 (5.3%)	4 (10.3%)	
Rhabdomyosarcoma	1 (1.3%)	2 (5.1%)	
MPNST	2 (3.3%)	0 (0.0%)	
Other	5 (6.6%)	1 (2.6%)	*p* = 0.01 *
**Neoadjuvant therapy**			
Nil	54 (71.1%)	37 (94.9%)	
Radiotherapy	1 (1.3%)	1 (2.6%)	
Chemotherapy	21 (27.6%)	1 (2.6%)	*p* = 0.001 *

* indicates statistical significance.

**Table 2 cancers-13-06284-t002:** Extended clinical information for patients receiving ICG pre-operatively.

#	Age	Sex	Location	Max Dimension (mm)	Histology	Grade	Neoadjuvant Therapy	Operating Consultants	ICG Dose	Time of Administration	Fluorescent	Surgical Guidance	Influence on Procedure	Margin
**1**	85	M	Thigh	65 mm	Myxofibrosarcoma	2	No	1	25 mg	16–24 h pre-op	No	No		Negative
**2**	65	M	Pelvis	125 mm	Chondrosarcoma	3	No	1, 2	75 mg	16–24 h pre-op	Yes	No		Negative
**3**	73	F	Forearm	85 mm	Myxofibrosarcoma	3	No	1	75 mg	16–24 h pre-op	Yes	Yes	Ligated the radial artery and took more tissue	Negative
**4**	78	M	Groin	130 mm	UPS	3	No	4, 8	75 mg	16–24 h pre-op	Yes	Yes	Identified a fluorescent lymph node with tumour	Negative
**5**	63	F	Upper arm	55 mm	Leiomyosarcoma	2	No	1	75 mg	16–24 h pre-op	Yes	No		Negative
**6**	75	F	Lower Leg	27 mm	Myxofibrosarcoma	3	No	2	75 mg	16–24 h pre-op	Yes	No		UPM
**7**	26	M	Chest wall	105 mm	Synovial sarcoma	3	No	4	75 mg	16–24 h pre-op	Yes	No		Negative
**8**	53	F	Tibia	114 mm	Osteosarcoma	High	Chemotherapy	1, 2, 8	75 mg	16–24 h pre-op	No	No		Negative
**9**	68	M	Thigh	38 mm	Myxofibrosarcoma	3	No	4	75 mg	16–24 h pre-op	Yes	No		Negative
**10**	55	F	Upper arm	80 mm	Pleomorphic rhabdomyosarcoma	3	No	1, 5	75 mg	16–24 h pre-op	Yes	Yes	More bicep taken to improve inferior margin	Negative
**11**	56	M	Proximal femur	105 mm	Chondrosarcoma	3	No	1	100 mg	16–24 h pre-op	Yes	No		Negative
**12**	56	F	Thigh	133 mm	UPS	3	No	3	100 mg	16–24 h pre-op	Yes	Yes	Assisted with perineural dissection	Negative
**13**	43	M	Shoulder	48 mm	Leiomyosarcoma	2	No	1	75 mg	16–24 h pre-op	Very weak	No		Negative
**14**	57	F	Thigh	87 mm	Synovial sarcoma	3	No	1	75 mg	16–24 h pre-op	Yes	Yes	Assisted with perivascular dissection	Negative
**15**	74	F	Thigh	220 mm	Myxofibrosarcoma	3	No	1, 2	75 mg	16–24 h pre-op	Yes	No		EPM
**16**	52	F	Pelvis	131 mm	Chondrosarcoma	3	No	1, 2, 6	75 mg	16–24 h pre-op	Yes	No		Negative
**17**	21	M	Pelvis	144 mm	Chondrosarcoma	3	No	2, 3	75 mg	16–24 h pre-op	Yes	No		Negative
**18**	76	M	Distal femur	250 mm	MPNST	High	No	2	75 mg	16–24 h pre-op	Yes	Yes	Surgeon changed the incision mark up due to the fluorescence	Negative
**19**	79	F	Knee	95 mm	Myxofibrosarcoma	3	No	1	50 mg	16–24 h pre-op	Yes	Yes	Surgeon changed the incision mark up due to the fluorescence	Negative
**20**	62	F	Thigh	72 mm	Myxofibrosarcoma	3	No	1	75 mg	16–24 h pre-op	Yes	No		Negative
**21**	55	M	Shoulder	175 mm	Chondrosarcoma	3	No	1, 2	75 mg	16–24 h pre-op	Yes	No		Negative
**22**	49	M	Thigh	90 mm	MPNST/dedifferentiated liposarcoma	3	No	1	75 mg	16–24 h pre-op	Yes	No		Negative
**23**	67	M	Humerus	170 mm	Chondrosarcoma	High	No	1	75 mg	16–24 h pre-op	Yes	Yes	Helped fashion the skin the fasciocutaneous flap	Negative
**24**	82	M	Femur	115 mm	Osteosarcoma	High	No	1, 2	75 mg	16–24 h pre-op	Yes	Yes	Assisted with perivascular dissection and posterior soft tissue margin	Negative
**25**	88	M	Thigh	250 mm	UPS	3	No	1, 2	75 mg	16–24 h pre-op	Yes	No		Negative
**26**	70	F	Ankle	45 mm	Myxofibrosarcoma	3	No	1, 2	75 mg	16–24 h pre-op	Yes	No		Negative
**27**	84	F	Buttock	125 mm	UPS	3	No	1	75 mg	16–24 h pre-op	Yes	No		Negative
**28**	69	M	Thigh	76 mm	Pleomorphic rhabdomyosarcoma	2	No	2	75 mg	16–24 h pre-op	Yes	No		Negative
**29**	73	M	Pelvis	105 mm	UPS	2	No	2, 3, 4	75 mg	16–24 h pre-op	Yes	No		UPM
**30**	81	M	Forearm	81 mm	Myxofibrosarcoma	2	Radiotherapy	1, 7	75 mg	16–24 h pre-op	Yes	No		Negative
**31**	88	M	Calf	140 mm	Leiomyosarcoma	3	No	1, 3	75 mg	16–24 h pre-op	Yes	No		EPM
**32**	73	F	Thigh	115 mm	Leiomyosarcoma	3	No	1, 2, 3, 4	75 mg	At induction	Yes	No		Negative
**33**	48	M	Pelvis	140 mm	Chondrosarcoma	3	No	2	100 mg	At induction	Yes	No		Negative
**34**	72	M	Pelvis	65 mm	Chondrosarcoma	3	No	1, 2	100 mg	At induction	Yes	No		Negative
**35**	62	F	Humerus	200 mm	Chondrosarcoma	2	No	1	75 mg	16–24 h pre-op	Yes	Yes	Assisted with perineural dissection	Negative
**36**	26	M	Hip	100 mm	Myxofibrosarcoma	3	No	1	75 mg	16–24 h pre-op	Yes	No		Negative
**37**	63	M	Thigh	105 mm	UPS	3	No	1, 2	1 mg/kg	16–24 h pre-op	Yes	Yes	Assisted with profunda femoris dissection	EPM
**38**	78	M	Shoulder	106 mm	UPS	3	No	1, 3	1 mg/kg	16–24 h pre-op	Yes	No		Negative
**39**	74	M	Trunk	105 mm	UPS	3	No	2, 4, 5	1 mg/kg	16–24 h pre-op	Yes	No		Negative

EPM = expected positive margin, UPM = unexpected positive margin.

**Table 3 cancers-13-06284-t003:** Margin status of all patients, split into those treated with conventional surgery and those receiving NIR guidance. UPM = unexpected positive margin, EPM = expected positive margin.

Variate	Margin Status		Total
	UPM (%)	EPM/Negative (%)	
**Conventional**	19 (25.0%)	57 (75.0%)	76
**NIR guided**	2 (5.1%)	37 (94.9%)	39
Total	21	94	115

**Table 4 cancers-13-06284-t004:** Univariate and multivariate analysis of the relationship between risk factors and unexpected positive margin rates.

Variate	Univariate		Multivariate	
	Exp(B)	*p*-Value	Exp(B)	*p*-Value
**Size**	2.568	0.230	9.18	**0.023**
**Site**	1.900	0.208	4.14	**0.033**
**Depth**	0.348	**0.032**	0.394	0.134
**Invasive phenotype**	5.394	**0.002**	6.74	**0.003**
**ICG guided**	0.162	**0.019**	0.100	**0.006**

## Data Availability

All necessary data is contained in the article.

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
