# Peer review of "Intraoperative Near-Infrared Fluorescence Guided Surgery Using Indocyanine Green (ICG) for the Resection of Sarcomas May Reduce the Positive Margin Rate: An Extended Case Series"

_cancers, 2021, doi:10.3390/cancers13246284_

Round 1

Reviewer 1 Report

A clearly written and rational manuscript that reports the findings of a study aimed at addressing an important clinical need – how best to improve the rate of adequate surgical clearance when resecting high grade soft tissue and bone sarcomas. The reported findings that the use of ICG/NIR was felt to provide helpful guidance to the surgeon in approximately a third of cases when used, and the markedly lower rates of unplanned involved surgical margins in patients where ICG/NIR, are of significant interest and provide a compelling case for prospective investigation, as suggested by the authors.

I have three major and several minor queries related to the study and the manuscript:

Major queries
1. Can the use of ICG/NIR to guide resection of sarcomas be regarded as a standard part of care? Should ICG be regarded as an investigational medicinal product in this context? Given that the use of ICG in this study seems to be outside its licensed indications, I feel this may be the case. If this is correct, the study ought to have been performed as a prospective CTIMP with associated requirement for MHRA and REC approval – no details of such approvals are provided. I would invite the authors to clarify and justify on what approval and consent basis this clinical research was performed.

  1. The key premise of the manuscript (that use of ICG and NIR helps to reduce rate of R1 resection) is predicated on the notion that the 2 patient groups (ICG/NIR vs control) are similar other than in the use of ICG. While the retrospective and non-randomised nature of the study will unavoidably confer limits to this, the authors should provide greater description of their methods and of patient characteristics to allow for assessment of to what degree various biases may have contributed to the observed association between use of ICG/NIR and favourable surgical outcome. This should include:
    - the criteria that informed use of ICG or not – the differences in baseline characteristics detailed in table 1 suggest that this was likely influenced by factors beyond those described (i.e the availability of the reagent and camera)
    - Who did the surgery and where? Was there an association between this and margin status and/or whether the surgeon found ICG/NIR to be helpful?
     - What governed choice of timing and dose of ICG administration? This should be further discussed given variation of ICG dose given as details in Table 2.
  2. Given the potential for ICG/NIR to guide surgeons to perform more extensive excisions than they would otherwise do, any difference in rate of surgical complication/morbidity would be an important endpoint for this and future studies. Inclusion of any available data on these outcomes would be welcome – if not available, some discussion of these considerations should be included.

Minor queries/comments

P2 para 1 line 7 – suggest ‘oncological outcome’ rather than survival

P2 para 2 line 5 – ‘when given prior to surgery’ – specify route of administration (not mentioned anywhere in manuscript – presumably IV?)

P2 para 2 – some discussion of mechanism for tumour-specific accumulation of ICG would be helpful (included in Discussion section but some discussion in introduction is warranted)

P3 para 1 line 2 – ref 23 does not seem to relate to R classification system. Need to specify distinction between R0 and R1 (e.g. tumour at margin or within 1mm?)

P4 para 1 line 7 – ‘invasive subtypes’ – this term is unclear, is it that ‘locally infiltrative’ is what is meant? If so, I am unaware of a rationale to include sarcoma NOS along with myxofibrosarcoma in this group – this requires referencing. Or is what is meant undifferentiated pleomorphic sarcoma, rather than sarcoma NOS?

P8 para line 3 – details of univariate and multivariate models should be provided in tabulated form wither as main table or supplemtary table

P11 – re: staged approach to surgery – this is an important point in the context of this trial which should be highlighted in the methods – in how many cases was further excision during index operation performed, and how was this informed by ICG/NIR?

Author Response

Thank you for taking the time to review our paper and for offering your incredibly helpful feedback to strengthen the paper. We have tried our upmost to accommodate all of your feedback as detailed below:

Major queries

  • COMMENT: Can the use of ICG/NIR to guide resection of sarcomas be regarded as a standard part of care? Should ICG be regarded as an investigational medicinal product in this context? Given that the use of ICG in this study seems to be outside its licensed indications, I feel this may be the case. If this is correct, the study ought to have been performed as a prospective CTIMP with associated requirement for MHRA and REC approval – no details of such approvals are provided. I would invite the authors to clarify and justify on what approval and consent basis this clinical research was performed.

REPLY: We appreciate your point r.e. the classification of ICG as a cTIMP. This has been discussed at length within the department and specialist teams during both this study and our current funding application for further study. After reviewing MHRA guidance, ICG is not a cTIMP as it is not itself a therapeutic. Given it is a widely used and well-established dye with a great deal of precedence in the literature for tumour identification, we felt that local approval for this study was appropriate. Patients were individually consented to the use of ICG prior to the procedure.

  • COMMENT: The key premise of the manuscript (that use of ICG and NIR helps to reduce rate of R1 resection) is predicated on the notion that the 2 patient groups (ICG/NIR vs control) are similar other than in the use of ICG. While the retrospective and non-randomised nature of the study will unavoidably confer limits to this, the authors should provide greater description of their methods and of patient characteristics to allow for assessment of to what degree various biases may have contributed to the observed association between use of ICG/NIR and favourable surgical outcome. This should include:

- the criteria that informed use of ICG or not – the differences in baseline characteristics detailed in table 1 suggest that this was likely influenced by factors beyond those described (i.e the availability of the reagent and camera)

- Who did the surgery and where? Was there an association between this and margin status and/or whether the surgeon found ICG/NIR to be helpful?

- What governed choice of timing and dose of ICG administration? This should be further discussed given variation of ICG dose given as details in Table 2.

REPLY: We have updated the methods to better describe this. Initially, one surgeon carried out the majority of the cases, prior to other surgeons in the department adopting this practice. There were no factors other than the availability of the camera and dye, as well as beds for pre-admission for administration of the dye, that influenced the choice of whether to use ICG. Whilst there are small differences between the groups, this will be largely due to the relatively modest sample size, particularly in the ICG arm.

  • COMMENT: Given the potential for ICG/NIR to guide surgeons to perform more extensive excisions than they would otherwise do, any difference in rate of surgical complication/morbidity would be an important endpoint for this and future studies. Inclusion of any available data on these outcomes would be welcome – if not available, some discussion of these considerations should be included.

REPLY: A very valid point – we have gone back through and collected wound complication data, defined this in the methods and discussed this in the results section. PROMs and functional outcomes were not available but we have added this in to the discussion now, along with the importance of collecting these data in future prospective studies.

Minor queries

  • COMMENT: P2 para 1 line 7 – suggest ‘oncological outcome’ rather than survival

REPLY: Changed as per recommendation.

  • COMMENT: .P2 para 2 line 5 – ‘when given prior to surgery’ – specify route of administration (not mentioned anywhere in manuscript – presumably IV?)

REPLY: Correct – we have updated this

  • COMMENT: P2 para 2 – some discussion of mechanism for tumour-specific accumulation of ICG would be helpful (included in Discussion section but some discussion in introduction is warranted)

REPLY: We have added a short explanation of this to the introduction

  • COMMENT: P3 para 1 line 2 – ref 23 does not seem to relate to R classification system. Need to specify distinction between R0 and R1 (e.g. tumour at margin or within 1mm?)

REPLY: R1 refers to tumour at the microscopic margin here (i.e. a margin of 0mm) – we have clarified this in the methods.

  • COMMENT: P4 para 1 line 7 – ‘invasive subtypes’ – this term is unclear, is it that ‘locally infiltrative’ is what is meant? If so, I am unaware of a rationale to include sarcoma NOS along with myxofibrosarcoma in this group – this requires referencing. Or is what is meant undifferentiated pleomorphic sarcoma, rather than sarcoma NOS?

REPLY: You are correct, we are referring to locally infiltrative and the correct nomenclature should be undifferentiated pleomorphic sarcoma rather than sarcoma NOS and we have changed this throughout accordingly.

  • COMMENT: P8 para line 3 – details of univariate and multivariate models should be provided in tabulated form wither as main table or supplementary table

REPLY: Table 4 has been added in to include these data.

  • COMMENT: P11 – re: staged approach to surgery – this is an important point in the context of this trial which should be highlighted in the methods – in how many cases was further excision during index operation performed, and how was this informed by ICG/NIR?

RESULTS: The staged approach refers specifically to leaving the wound open until the margins are available (~1 week) prior to either further resection or reconstruction depending on the results. With regards to further excision within the index operation using ICG, we have added information to the table to explain exactly how the surgeon felt that ICG guided the procedure.

Reviewer 2 Report

In this retrospective study, the authors evaluated efficacy of using ICG fluorescence imaging for intraoperative identification of sarcomas. Between 39 patients undergoing resection with ICG fluorescence guidance and 76 resected cases without a use of fluorescence imaging, the former group showed a significantly lower incidence of unexpected positive margin (5.1% vs. 25%). This study was not a prospective trial but would be one of the largest series of sarcoma surgery with fluorescence guidance, including valuable information on the clinical application of ICG fluorescence imaging to this specific surgical field.

  1. Please describe settings for intraoperative fluorescence imaging in more detail (i.e. timing, distance from camera head, mode [monochromatic images/SPY mode/superimposed images], on/off status of operation/ceiling lights) for readers to reproduce the authors’ techniques.

  1. Were there any overlaps in operating surgeons between the groups? This factor might affect the outcomes of surgical margins and surgeons’ comments on the impact of fluorescence imaging on surgical decision making.

  1. Dose of ICG seems to be markedly high as compared with that used for identification of the other tumors. For example, in the field of liver surgery where ICG has been applied more commonly, 0.5 mg/kg of ICG is used both for tumor imaging and liver function test (PMID: 19326450). Although mechanistic background of liver cancer identification by ICG fluorescence imaging (biliary excretion disorders) is quite different from situation in the other tumors including sarcoma (mainly EPR effect), the possibility of reducing ICG dose should further be discussed.

  1. Tumor identification by fluorescence imaging would potentially be more effective in a case of re-operation for recurrences because of adhesions around cancerous tissues. Did this series include any cases of recurrence? If the authors excluded any patients with special conditions like recurrence, please clarify the inclusion and exclusion criteria for this study.

  1. In how many cases were the residual tumorous tissues identified and removed by ICG fluorescence imaging on raw surfaces of resection?

  1. Did the authors microscopically evaluate specific sites of accumulation of ICG in sarcoma tissues? With a use of recent NIR fluorescence microscopy, fluorescence signals emitted from ICG could be detected and overlapped on H&E staining of tissue samples.

Author Response

Thank you for taking the time to review our paper and for offering your incredibly helpful feedback to strengthen the paper. We have tried our upmost to accommodate all of your feedback as detailed below:

  • COMMENT: Please describe settings for intraoperative fluorescence imaging in more detail (i.e. timing, distance from camera head, mode [monochromatic images/SPY mode/superimposed images], on/off status of operation/ceiling lights) for readers to reproduce the authors’ techniques.

REPLY: Excellent point – we have added further detail in the methods (section 2.2) as to the exact procedure used. There was no set distance between the camera and tumour, we just aimed to image the of interest in the frame (approximately 30-50cm).

  • COMMENT: Were there any overlaps in operating surgeons between the groups? This factor might affect the outcomes of surgical margins and surgeons’ comments on the impact of fluorescence imaging on surgical decision making.

REPLY: The same surgeons performed the procedures in both groups and we have added the surgeons involved into Table 2. As multiple consultants were often involved in the same procedure, we have not been able to analyse the influence of this on margins, or whether it was felt to influence the procedure. We have also included information into Table 2 as to how they felt it influenced the procedure, although we appreciate this will always be subjective.

  • COMMENT: Dose of ICG seems to be markedly high as compared with that used for identification of the other tumors. For example, in the field of liver surgery where ICG has been applied more commonly, 0.5 mg/kg of ICG is used both for tumor imaging and liver function test (PMID: 19326450). Although mechanistic background of liver cancer identification by ICG fluorescence imaging (biliary excretion disorders) is quite different from situation in the other tumors including sarcoma (mainly EPR effect), the possibility of reducing ICG dose should further be discussed.

REPLY: This is a very reasonable suggestion and we have added this to our discussion. As you say, the mechanism for liver cancer is different, with the hepatic excretion of ICG likely resulting in higher concentrations accumulating in the liver. In addition, ICG has a good safety profile and we are still using is well below the maximum dose of 2mg/kg. We agree that this should be explored in the future however.

  • COMMENT: Tumor identification by fluorescence imaging would potentially be more effective in a case of re-operation for recurrences because of adhesions around cancerous tissues. Did this series include any cases of recurrence? If the authors excluded any patients with special conditions like recurrence, please clarify the inclusion and exclusion criteria for this study.

REPLY: We did not include any recurrences in this series but that is something we will do going forwards. The inclusion criteria is described in the first paragraph of section 2.1.

  • COMMENT: In how many cases were the residual tumorous tissues identified and removed by ICG fluorescence imaging on raw surfaces of resection?

REPLY: This is difficult to answer; the images often resulted in changing the margin pre-resection rather than from the wound bed which makes it difficult to determine whether or not it directly led to avoiding an unexpected positive margin. We have added a column to table 2 to describe how the images influenced the surgeon’s decision making however.

  • COMMENT: Did the authors microscopically evaluate specific sites of accumulation of ICG in sarcoma tissues? With a use of recent NIR fluorescence microscopy, fluorescence signals emitted from ICG could be detected and overlapped on H&E staining of tissue samples.

REPLY: Great question – yes we have, although this will be published in the future as part of a separate basic science paper.

Reviewer 3 Report

Please address these questions:

1. In section 1, page 2, there should be an introduction to the mechanism of NIF guided surgery using ICG and explain how ICG is specific to the tumor.

2. In section 3.2, page 4, “In 11/39 cases, the surgeon felt that the ICG fluorescence helped guide their intra-operative decision making with regards to margins taken. NIR fluorescence-guided resections were carried out by 8 different surgeons;…”. Is there a standard of deciding the “Surgical Guidance”? The 8 surgeons may have different experiences in imaging-guided surgery; hence their reports of surgical guidance are likely to be subjective to the individual surgeon. This concern should be addressed. In table 2, the surgeon who did the surgery in each case should be listed (e.g. Surgeon 1, Surgeon 2…)

3. In table 2, page 6, 4 out of 5 positive margins happened to the patients with lower extremity sarcomas. Is the margin related to the location of the tumor since the different structures of tumor location may give different difficulties in surgery?

4. In section 3.3, page 8, “This retains significance when tumor size (<50mm vs ≥50mm), site (extremity vs non-extremity) and invasive phenotype (myxofibrosarcoma/sarcoma NOS vs. other sarcomas), are included as a multivariate model (p=0.007)”. How was the multivariate model selected? Why is age not included? Each should be explained (i.e., why 50mm, extremity, myxofibrosarcoma/sarcoma are selected).

5. In table 3, page 8, what is the EPM number and percentage in conventional surgery cases? The EPM should not be mixed with negative cases, and it should be omitted from all the margin statuses. According to my understanding, the EPMs are expected regardless of the method of surgery.

6. In section 4, page 10, “Furthermore, we demonstrated that ICG guided surgery is feasible in patients receiving radiotherapy in the neoadjuvant setting.” Only one patient received radiotherapy after the ICG guided surgery, which is insufficient to support the conclusion “ICG guided surgery is feasible in patients receiving radiotherapy in the neoadjuvant setting”.

Author Response

Thank you for taking the time to review our paper and for offering your incredibly helpful feedback to strengthen the paper. We have tried our upmost to accommodate all of your feedback as detailed below:

  • COMMENT: In section 1, page 2, there should be an introduction to the mechanism of NIF guided surgery using ICG and explain how ICG is specific to the tumor.

REPLY: Excellent suggestion, we have added this in.

  • COMMENT: In section 3.2, page 4, “In 11/39 cases, the surgeon felt that the ICG fluorescence helped guide their intra-operative decision making with regards to margins taken. NIR fluorescence-guided resections were carried out by 8 different surgeons; ”. Is there a standard of deciding the “Surgical Guidance”? The 8 surgeons may have different experiences in imaging-guided surgery; hence their reports of surgical guidance are likely to be subjective to the individual surgeon. This concern should be addressed. In table 2, the surgeon who did the surgery in each case should be listed (e.g. Surgeon 1, Surgeon 2…)

REPLY: The same surgeons performed the procedures in both groups and we have added the surgeons involved into Table 2. As multiple consultants were often involved in the same procedure, we have not been able to analyse the influence of this on margins, or whether it was felt to influence the procedure. We have also included information into Table 2 as to how they felt it influenced the procedure, although we appreciate this will always be subjective.

  • COMMENT: In table 2, page 6, 4 out of 5 positive margins happened to the patients with lower extremity sarcomas. Is the margin related to the location of the tumor since the different structures of tumor location may give different difficulties in surgery?

REPLY: For patient 29 (table 2), the UPM may well be location related as this was a difficult pelvic tumour. Otherwise however, we think this is just related to the small series and the infiltrative nature of some sarcomas.

  • COMMENT: In section 3.3, page 8, “This retains significance when tumor size (<50mm vs ≥50mm), site (extremity vs non-extremity) and invasive phenotype (myxofibrosarcoma/sarcoma NOS vs. other sarcomas), are included as a multivariate model (p=0.007)”. How was the multivariate model selected? Why is age not included? Each should be explained (i.e., why 50mm, extremity, myxofibrosarcoma/sarcoma are selected).

REPLY: We have added an explanation of this, along with references to section 2.3. We selected variables that either had associations with positive margins or local recurrence within the literature (as referenced in the paper). Age was not included as this is not associated with either of these as far as we are aware.

  • COMMENT: In table 3, page 8, what is the EPM number and percentage in conventional surgery cases? The EPM should not be mixed with negative cases, and it should be omitted from all the margin statuses. According to my understanding, the EPMs are expected regardless of the method of surgery.

REPLY: EPM only applies to the very specific margin bordering with the critical structure. If a different margin was returned positive, we would class this a UPM. As such, we have included EPM and negative margins together. Furthermore, previous research has shown that the LR rate between negative and EPM against critical structures is the same (PMID: 24894656)

  • COMMENT: In section 4, page 10, “Furthermore, we demonstrated that ICG guided surgery is feasible in patients receiving radiotherapy in the neoadjuvant setting.” Only one patient received radiotherapy after the ICG guided surgery, which is insufficient to support the conclusion “ICG guided surgery is feasible in patients receiving radiotherapy in the neoadjuvant setting”.

REPLY: We have changed the wording to ”may be feasible” to accommodate this.

Round 2

Reviewer 1 Report

Thank you for addressing my comments in your revised manuscript - I feel that these have now been addressed satisfactorily 

Reviewer 3 Report

They addressed all my concern and questions, I would like to suggest to accept this paper as current version.